# Exploring Genetic and Epigenetic Changes in Lingonberry Using Molecular Markers: Implications for Clonal Propagation

Umanath Sharma [1,2], Arindam Sikdar [1,2], Abir U. Igamberdiev [1] and Samir C. Debnath [2,*]

1   Department of Biology, Memorial University of Newfoundland, 45 Arctic Avenue, St. John's, NL A1C 5S7, Canada; usharma@mun.ca (U.S.); asikdar@mun.ca (A.S.); igamberdiev@mun.ca (A.U.I.)
2   St. John's Research and Development Centre, Agriculture and Agri-Food Canada, 204 Brookfield Road, St. John's, NL A1E 0B2, Canada
*   Correspondence: samir.debnath@agr.gc.ca

**Abstract:** Lingonberry (*Vaccinium vitis-idaea* L.) is an important and valuable horticultural crop due to its high antioxidant properties. Plant tissue culture is an advanced propagation system employed in horticultural crops. However, the progeny derived using this technique may not be true-to-type. In order to obtain the maximum return of any agricultural enterprise, uniformity of planting materials is necessary, which sometimes is not achieved due to genetic and epigenetic instabilities under in vitro culture. Therefore, we analyzed morphological traits and genetic and epigenetic variations under tissue-culture and greenhouse conditions in lingonberry using molecular markers. Leaf length and leaf width under greenhouse conditions and shoot number per explant, shoot height and shoot vigor under in vitro conditions were higher in hybrid H1 compared to the cultivar Erntedank. Clonal fidelity study using one expressed sequence tag (EST)—polymerase chain reaction (PCR), five EST—simple sequence repeat (SSR) and six genomic (G)—SSR markers revealed monomorphic bands in micropropagated shoots and plants in lingonberry hybrid H1 and cultivar Erntedank conforming genetic integrity. Epigenetic variation was studied by quantifying cytosine methylation using a methylation-sensitive amplification polymorphism (MSAP) technique. DNA methylation ranged from 32% in greenhouse-grown hybrid H1 to 44% in cultivar Erntedank under a tissue culture system. Although total methylation was higher in in vitro grown shoots, fully methylated bands were observed more in the greenhouse-grown plants. On the contrary, hemimethylated DNA bands were more prominent in tissue culture conditions as compared to the greenhouse-grown plants. The study conclude that lingonberry maintains its genetic integrity but undergoes variable epigenetic changes during in vitro and ex vitro conditions.

**Keywords:** DNA methylation; epigenetic variation; greenhouse-grown plants; in vitro culture; molecular markers; shoot proliferation

## 1. Introduction

Plants respond to changes in the environment by altering their growth, physiology and reproductive processes. The molecular basis of such changes is based on the alteration in the underlying DNA or its plastic modification, including DNA methylation [1]. Commercial micropropagation is performed in a unique optimized environment containing various inorganic nutrients and growth hormones, controlled light, humidity and osmotic conditions. However, such an artificial environment may create a stressful situation for the plant material, resulting in genetic or epigenetic changes [2]. Among epigenetic mechanisms, DNA methylation is the most important phenomenon affecting plant phenotype [3]. However, for the sake of uniformity, both genetic and epigenetic variations are not desired in the commercial micropropagation of plants, including that of lingonberry (*Vaccinium vitis-idaea* L., family *Ericaceae*).

Lingonberry is a berry fruit bearing shrub commonly found in the Northern Hemisphere [4,5]. The lingonberry plant bears red edible fruits; both fruits and leaves are rich

in nutrients and bioactive compounds, such as sugars, organic acids, vitamins, minerals, dietary fiber, and polyphenolics [4,6]. While lingonberries are cultivated in some parts of the world, the majority of the fruits are collected from wild natural habitats and are consumed fresh and frozen. A substantial part of a harvest is processed into food products bearing longer shelf-life and pharmaceutical products [4]. Lingonberry possesses strong antiviral, antimicrobial, antioxidant, anti-inflammatory, and neuroprotective potential [7]. Therefore, eating fresh or processed lingonberries may reduce the risk or eliminate the development of gastrointestinal, metabolic, cardiovascular, renal, and neurodegenerative disorders [8].

The propagation of lingonberry is generally performed by vegetative methods using the rhizome because, being genetically heterozygous, progeny derived from lingonberry seeds are not true-to-type. Although vegetative propagation retains the genetic characteristics, this method is not economically viable in lingonberry due to its short life span and poor rhizome production [9]. Commercial production requires a large number of uniform plants, which can be achieved using in vitro propagation techniques. Propagation of lingonberry by tissue culture is much faster than traditional methods [10], but occasional variations in the tissue-cultured progeny, termed somaclonal variations, have been reported in several crops [11]. Therefore, we assessed the genetic and epigenetic stability of lingonberry in tissue culture medium. Genetic variation in tissue culture can arise due to point mutations, chromosomal rearrangements, relocation of mobile genetic elements, or changes in the ploidy level [12]. Although epigenetic mechanisms stabilize cell identity and maintain tissue organization, they entail a variety of reversible biochemical modifications that can occur on the underlying DNA, its interacting proteins, or both, modifying chromatin structure and resulting in an altered phenotype [13]. At the molecular level, such epigenetic phenomena are moderated by reversible mechanisms such as histone modifications, DNA methylation, and small RNAs, thus affecting the regulatory states of genes [13,14].

In plants, DNA methylation of cytosine base is a widespread epigenetic mechanism that contributes to the regulation of gene expression, maintenance of genomic integrity, cellular differentiation, and plant response to biotic and abiotic stresses [3,13,15]. DNA methylation also plays a vital role in many crucial biological processes, such as genomic imprinting, transposable element silencing, maintenance of heterochromatin, and inactivation of X-chromosome [16]. DNA methylation occurs when a methyl group is transferred from *S*-adenosyl methionine (SAM) to the fifth carbon of cytosine residue of DNA by DNA methyltransferases [1]. In plants, DNA methylation occurs at symmetric mCG and mCHG, or asymmetric mCHH contexts, where mC = methylated cytosine, and H = A, T or C [17]. In *Arabidopsis thaliana*, 55% of methylated cytosines are reported in CG sites, while cytosine methylation accounts for 23% and 22% at CHG and CHH sites, respectively [18]. The gene's function is reported to be affected by the position where the methylation of cytosines has occurred, for example, in the regions of transposons or the promoter regions of the gene [19]. Because of the heritable nature of variation, DNA methylation marks are useful in sexually and asexually propagated crops. Given the significant role that DNA methylation has in the regulation of gene expression [20], it is relevant to investigate how different growth conditions affect cytosine methylation.

Methylation-sensitive amplification polymorphism (MSAP) is a modified amplified fragment length polymorphism (AFLP) method commonly used to study DNA cytosine methylation [21,22]. In the place of the use of a frequent-cutter restriction enzyme, *Mse*I, in the AFLP technique, DNA is cleaved using two different enzymes, *Hpa*II and *Msp*I. The recognition sequence for both of these restriction enzymes is CCGG; however, they cleave the DNA fragment based on the particular pattern of methylated cytosines. This method is popular as it offers several advantages, especially in non-model plants such as lingonberry: because the obtained loci cover the information on the whole genome, obtaining a general idea about the DNA methylation is relatively quick, and the method is cost-effective compared to other techniques such as whole genome bisulfite sequencing [21].

For commercial micropropagation, genetic and epigenetic stability is necessary for phenotypic integrity. As a super food, lingonberry is increasing in popularity day by day and, therefore, holds huge potential for commercialization. In this context, we studied the clonal fidelity and global DNA methylation in micropropagated and greenhouse-grown lingonberry leaves using molecular markers. Information obtained through this investigation is expected to contribute to the commercialization of lingonberry as a medicinally important crop.

## 2. Materials and Methods

### 2.1. Plant Material, Growth Conditions and Morphological Data

Lingonberry cultivar Erntedank and a selected hybrid designated as H1, developed at St. John's Research and Development Centre, Agriculture and Agri-Food Canada, St. John's, Newfoundland and Labrador, Canada [6], were used for this study. Ten plants from each genotype were used to record the morphological data. For the genetic analysis, five plants were randomly selected, chopped, mixed and sampled for DNA analysis. Each experiment was replicated three times.

In vitro cultures were initiated using transversely segmented leaf explants from cultivar Erntedank and hybrid H1 following Arigundam et al. [10]. Surface sterilized explants were inoculated on a semi-solid medium on Fisherbrand™ Petri dishes covered with clear lids. The 25 cm$^3$ sterilized basal medium in each Petri dish contained 3/4 micro salts and macro salts [10] supplemented with 20 g dm$^{-3}$ sucrose, 1.25 g dm$^{-3}$ gelrite and 3.5 g dm$^{-3}$ Sigma A 1296 agar, pH 5.0. The plant growth regulator (PGR) added to the medium was zeatin at a concentration of 1 mg L$^{-1}$. The cultures were kept in dark in a growth chamber at $20 \pm 2\ ^\circ$C and relative humidity of 60–70%. After 2 weeks, the cultures were then exposed to cool white fluorescent lamps emitting PPFD of 30 µmol m$^{-2}$ s$^{-1}$. After 4 weeks of inoculation, culture-initiated explants were transferred to Sigma bottles containing the same medium [10] where shoot regeneration was obtained. Leaf length and leaf width measurements were taken from 2-year-old greenhouse-grown lingonberry genotypes. Other morphological data were taken from 2-month-old cultures of cultivar Erntedank (Figure 1) and hybrid H1. Leaf length, leaf width and shoot height were measured in cm. The shoots per explant and the leaves per shoot were counted, and shoot vigor was assessed using the visual scale of 1–8, 1 being the poorest and 8 being the best looking shoot [10]. The shoots were then sampled for DNA analysis.

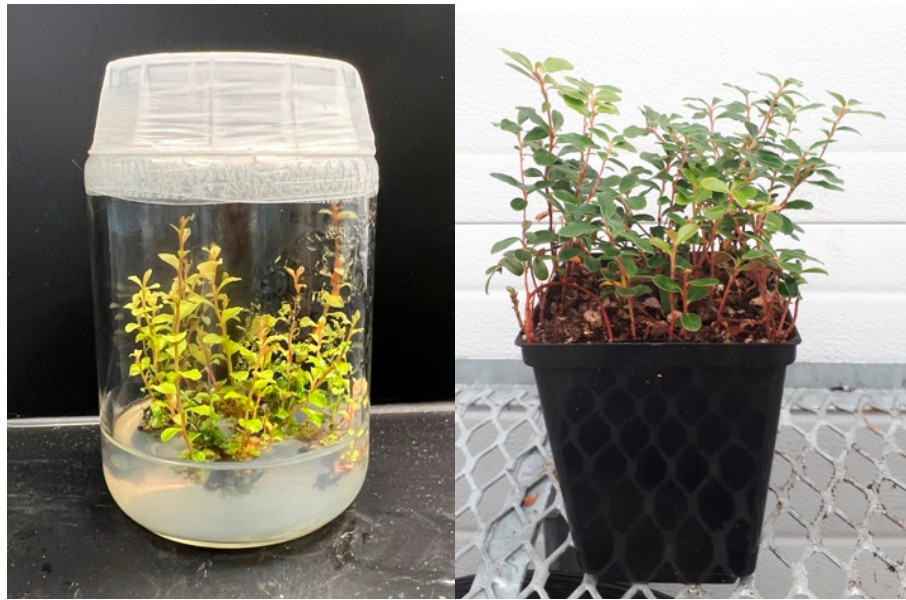

**Figure 1.** Two-month-old shoots of lingonberry cultivar Erntedank in in vitro semi-solid medium in a Sigma bottle (**left**) and greenhouse-grown plants in a plastic pot (**right**) containing peat and perlite medium.

From mother plants, young leaves were taken for DNA extraction from the cultivar Erntedank (Figure 1) and hybrid plants that were maintained in a greenhouse under natural light conditions having photosynthetic photon flux density (PPFD) of 90 μmol m$^{-2}$ s$^{-1}$, temperature of 20 ± 2 °C and relative humidity of 85% maintained using automatic control systems. Plants were grown and maintained in 10 cm plastic pots containing peat and perlite in the ratio of 2:1 (*v/v*).

### 2.2. DNA Isolation

Actively growing lingonberry leaves from greenhouse plants and shoots from tissue culture plants (100 mg) were sampled, and genomic DNA was isolated using DNeasy Plant Mini Kits (Qiagen GmbH, Hilden, Germany) following the manufacturer's instructions. Briefly, the sampled lingonberry leaves were shock-frozen in liquid nitrogen immediately after collection and stored at −80 °C in a freezer until DNA isolation. Lysis was performed in 2 mL centrifuge tubes containing 600 μL AP1 buffer (80% ethanol, 100 mM NaCl and 10 mM Tris-HCl, pH 7.5) using two ceramic beads in a FastPrep 24 tissue and cell homogenizer (MP Biomedicals, Irvine, CA, USA). Then, 20 μL proteinase K (20 mg/mL) was added to the mixture and incubated for 1 h at 65 °C. Subsequently, 4.5 μL RNAse (100 mg/mL) was added to the mix and incubated for 15 min under the same conditions at 65 °C. Neutralization was performed using 425 μL P3 buffer, incubated at −20 °C for 18 min. DNA was separated from the mixture using a QIAshredder Mini Spin column placed in a 2 mL collection tube and centrifuged at 20,000× *g*. The lysate was washed using 1.5 volume of AW1 and 500 μL AW2 in a DNeasy mini spin column, where the DNA was trapped in the DNeasy membrane, which was eluted using 50 μL AE buffer. The DNA concentration of 50 ng/μL was maintained across the samples using 1 × TE buffer. The quality of the DNA was assessed using the absorbance ratio of A260 to A280 in the range of 1.8–1.9 and absorbance ratio of A260 to A230 in the range of 2.0–2.2 using a nanodrop spectrophotometer. The DNA was used to assess global DNA methylation in tissue culture and greenhouse-grown plants using the methylation-sensitive amplification polymorphism technique.

### 2.3. Clonal Fidelity Experiment

DNA samples from cv. Erntedank were diluted to 10 ng μL$^{-1}$ using 1 × TE buffer. Amplification of the DNA regions containing markers was carried out using 12 molecular markers including one expressed sequence tag (EST)—PCR, five EST—simple sequence repeats (SSR) and six genomic simple sequence repeats (GSSR) [23] that were proven effective on *Vaccinium* species. Amplification reactions were carried out in a 25 μL reaction mixture containing 2.5 μL DNA template (25 ng DNA per reaction), 2.5 μL PCR buffer (1.5 mM MgCl$_2$), 0.1 μL Taq DNA polymerase (5U μL$^{-1}$ stock), 0.5 μL dNTP (10 mM stock), and 0.5 μL primer (10 μL stock), and the final volume was adjusted with PCR-grade water (Sigma Chemical Co., St. Louis, MO, USA). DNA amplification was performed in Mastercycler ep Gradient S (Eppendorf AG, 22331 Hamburg, Germany). Initial denaturation was carried out at 94 °C for 10 min. The reaction was run for 40 cycles. Each cycle of amplification reaction consisted of denaturation of template DNA at 92 °C for 40 s. Primer extension was attained at 72 °C for 2 min. The reaction was completed with the final extension allowed to incubate at 72 °C for 10 min.

Separation of DNA fragments was performed by gel electrophoresis in 1.6% agarose (Agarose 3:1 HRB™, Amresco, Solon, OH, USA) gel pre-casted in a solution containing Tris-borate EDTA buffer (TBE) and GelRed nucleic acid stain (Biotium Inc., Hayward, CA, USA) in the ratio of 2:1.

A 100 bp Low-Ranger and a 50 bp Mini sizer DNA ladder (Norgen Bioteck Corp., Thorold, ON, Canada) was used as size marker. The gel was run for 1.2 h at 100 V. DNA bands were photographed digitally under UV light using a gel documentation system (InGenius 3; Syngene, Beacon House, Cambridge, UK). Visual observation of presence or absence of bands was recorded for further interpretation.

### 2.4. Methylation-Sensitive Amplification Polymorphism (MSAP) Assay

DNA samples from the cultivar Erntedank and a hybrid lingonberry were used to determine cytosine methylation. The AFLP technique for DNA fingerprinting [24], modified to the MSAP technique by [21], was adopted in the experiment. The MSAP assay was performed in the following steps, using MSAP adapters and primers (Table 1).

**Table 1.** List of adapter sequences, preamplification primers and selective amplification primers for methylation-sensitive amplification polymorphism analysis of in vitro-propagated and greenhouse-grown lingonberry genotypes.

| Oligo Name | Function | Nucleotide Sequences |
|---|---|---|
| Ad. *Eco*RI | Forward adaptor | 5′-CTG TAG ACT GCG TAC C-3′ |
| Ad. *Eco*RI | Reverse adaptor | 3′-CAT CTG ACG CAT GGT TAA-5′ |
| Ad. *Msp*I/*Hpa*II | Forward adaptor | 5′-GAT CAT GAG TCC TGC T-3′ |
| Ad. *Msp*I/*Hpa*II | Reverse adaptor | 3′-AGT ACT CAG GAC GAG C-5′ |
| *Eco*RI (E) | Preselective amplification primer | 5′-GAC TGC GTA CCA ATT CA-3′ |
| *Msp*I/*Hpa*II (MH) | Preselective amplification primer | 5′-ATC ATG AGT CCT GCT CGG-3′ |
| E-TT | Selective amplification primer | 5′-GAC TGC GTA CCA ATT CAT T-3′ |
| E-TG | Selective amplification primer | 5′-GAC TGC GTA CCA ATT CAT G-3′ |
| MH-ATG | Selective amplification primer | 5′-ATC ATG AGT CCT GCT CGG ATG-3′ |
| MH-AAC | Selective amplification primer | 5′-ATC ATG AGT CCT GCT CGG AAC-3′ |
| MH-AAG | Selective amplification primer | 5′-ATC ATG AGT CCT GCT CGG AAG-3′ |
| MH-ACA | Selective amplification primer | 5′-ATC ATG AGT CCT GCT CGG ACA-3′ |
| MH-ATT | Selective amplification primer | 5′-ATC ATG AGT CCT GCT CGG ATT-3′ |
| MH-TCC | Selective amplification primer | 5′-ATC ATG AGT CCT GCT CGG TCC-3′ |
| MH-AAT | Selective amplification primer | 5′-ATC ATG AGT CCT GCT CGG AAT-3′ |
| MH-TCG | Selective amplification primer | 5′-ATC ATG AGT CCT GCT CGG TCG-3′ |

#### 2.4.1. Digestion

Digestion of DNA (900–1100 ng) was performed with *Eco*RI, *Msp*I, and *Hpa*II restriction endonuclease (Thermo Scientific, Waltham, MA, USA). Genomic DNA in a 75 μL reaction volume containing 3× FastDigest buffer was cleaved using 3 U of *Eco*RI at 37 °C for 1.5 h. Inactivating *Eco*RI, the reaction was stopped by incubating the mixture at 65 °C for 10 min. The *Eco*RI-digested DNA was allocated into three distinct aliquots and subjected to three separate reactions in a 50 μL reaction volume containing 1× the corresponding buffer. One of the aliquots was digested with 2 U *Msp*I, another with 2 U *Hpa*II, and the remaining aliquot was treated with 2 U each of *Msp*I and *Hpa*II restriction endonucleases and incubated for 3 h at 37 °C. The restriction enzyme reaction were stopped by denaturing the enzymes and incubating for 15 min at 65 °C.

#### 2.4.2. Ligation

The digested DNA (50 μL) was ligated to adapters using 5 U of T4 DNA ligase, 10 μL 1 × T4 DNA ligase buffer, 1 μL of 10 μM *Eco*RI adapter, 1 μL of 100 μM *Msp*I/*Hpa*II adapter and 2 μL polyethylene glycol (50% *w/v*). The final volume was adjusted to 100 μL using PCR water and incubated at 23 °C for 5 h. The reaction of the enzymes was stopped by placing the mixture at 65 °C for 10 min.

#### 2.4.3. Preamplification

The DNA fragments (4 μL) ligated to the adaptors were amplified by PCR using *Eco*RI (E) as forward and *Msp*I-*Hpa*II (MH) as reverse primers (Table 1). A total volume of 50 μL pre-selective amplification was carried out containing a final concentration of 200 μM of each dNTP (Amresco LLC, Solon, OH, USA), 1× PCR buffer (Qiagen Inc., Toronto, ON, Canada), 1 U of Top Taq DNA polymerase (Qiagen) and 0.2 μM of E and MH primers. PCR amplifications were carried out in an Eppendorf Mastercycler Gradient thermocycler (Eppendorf AG, 22331 Hamburg, Germany). Pre-selective amplification products showed a 100 to 1000 bp smear in 1.8% agarose gel. PCR products were diluted seven times using 0.1 × TE buffer for selective amplification.

### 2.4.4. Selective Amplification

Diluted pre-amplified products from the previous step were selectively amplified with 16 primer combinations in total. The two *EcoR*I selective amplification primers consisted of two particular extra bases (TT and TG) of preamplification *EcoR*I (E) as forward primers, and eight *Msp*I-*Hpa*II primers consisting of three different base overhangs (ATG, AAC, AAG, ACA, ATT, TCC, AAT, TCG) of *Msp*I-*Hpa*II (MH) preamplification primers used as reverse primers. A total volume of 25 μL PCR amplification reaction contained 1× PCR buffer, 400 μM dNTPs, 0.4 μM of each selective primer, 1 U of Top Taq DNA polymerase, and 4 μL pre-amplified PCR product. PCR amplification was performed using the touch-down cycles with the following conditions: initially heated to 94 °C for 5 min; then, 13 cycles were run for 30 s at 94 °C, 1 min at 65 °C, which was reduced by 0.7 °C per cycle, and 72 °C for 2 min, followed by 23 cycles of 30 s at 94 °C, 1 min at 56 °C and 2 min at 72 °C with a final extension step of 72 °C for 10 min. The selective amplification products were separated on 6% denaturing polyacrylamide gel electrophoresis (PAGE).

### 2.4.5. Polyacrylamide Gel Electrophoresis (PAGE)

Denaturing formamide dye solution was prepared using 10 mM EDTA of pH 8.0, 98% formamide, 0.01% (*w/v*) xylene cyanol and 0.01% (*w/v*) bromophenol blue. Selective amplification products were denatured by mixing them with equal volumes of denaturing formamide dye and heating for 4 min at 95 °C, then cooling immediately for 5 min and keeping at −20 °C. The 6% PAGE gels were pre-run at 90 V for about 1 h to clean the wells. Denatured selective amplification products (10 μL) were loaded in the gels, and a potential difference of 95 V was applied for 3 h. The DNA fragments separated in gels were stained for 30 min in the dark in 1% PAGE GelRed™ (Biotium Inc., Hayward, CA, USA) with gentle agitation and visualized using InGenius 3 gel documentation system (Syngene, Frederick, MD, USA). The 1 kb and 50 bp DNA ladders (Norgen Biotek, Thorold, ON, Canada) were used as a molecular size marker. The experiments were repeated twice, and the reproducible results were used to score for further analysis.

### 2.4.6. Profiling Scoring and Data Analysis

The methylation status at tetranucleotide restriction sites (5′-CCGG-3′) was detected by comparing the DNA profiles, based on the presence or absence of DNA bands by the reaction of restriction enzymes *EcoR*I+*Msp*I, *EcoR*I+*Hpa*II and *EcoR*I+*Msp*I+*Hpa*II. In the absence of methylation at 5′-CCGG-3′, both the isoschizomers *Msp*I and *Hpa*II cleave the DNA fragments at this site. Therefore, the DNA bands present in all three lanes were considered non-methylation (first case). DNA bands identified in *EcoR*I+*Msp*I and *EcoR*I+*Msp*I+*Hpa*II lanes but absent from the *EcoR*I+*Hpa*II lane were considered as fully methylated internal cytosine, as *Hpa*II cannot cleave at fully methylated internal cytosine (second case). The bands that were present in the lanes of both *EcoR*I+*Hpa*II and *EcoR*I+*Msp*I+*Hpa*II but not in *EcoR*I+*Msp*I were considered hemimethylation of external cytosine (5′-mCCGG-3′, third case). In another case, bands absent from the *EcoR*I+*Msp*I+*Hpa*II lane but present in the *EcoR*I+*Hpa*II lane indicated the existence of a site for internal 5′-CmCGG-3′ [25] and were scored accordingly (fourth case). In aggregate, the number of bands present in the second, third, and fourth cases resulted in the total number of methylated bands.

$$\text{Total methylation} = \frac{\text{Methylated band numbers}}{\text{Total band numbers}} \times 100\% \tag{1}$$

$$\text{Fully methylated percentage} = \frac{\text{Fully methylated band numbers}}{\text{Total methylated band numbers}} \times 100\% \tag{2}$$

$$\text{Hemimethylated percentage} = \frac{\text{Hemimethylated band numbers}}{\text{Total methylated band numbers}} \times 100\% \tag{3}$$

### 2.4.7. Statistical Analysis

Morphological statistical data analysis was performed through one-way ANOVA in jamovi [26] software (3rd generation) followed by standard error (SE) and mean.

## 3. Results

### 3.1. Morphological Characteristics

Among the greenhouse-grown lingonberry samples, the average leaf length of hybrid lingonberry H1 was 2.44 cm; this was significantly longer than that of the lingonberry cultivar, which was only 1.98 cm (Figure 2). Similarly, the average leaf width of hybrid lingonberry was 1.3 cm, which was significantly wider than that of the cultivar, for which the average width of leaf was 1.04 cm under greenhouse conditions. In tissue cultures, the average shoots per explant in hybrid was 4, which was significantly more than that for the cultivar with an average of 3 shoots per explant. Similarly, shoot height was significantly higher in the hybrid (3.67 cm) as compared to the cultivar (3.36 cm). Plants of hybrid lingonberry showed significantly better vigor (average 6.4) as compared to the cultivar (average 5.9). However, there was no any significant difference in the number of shoots between the studied lingonberry cultivar and the hybrid.

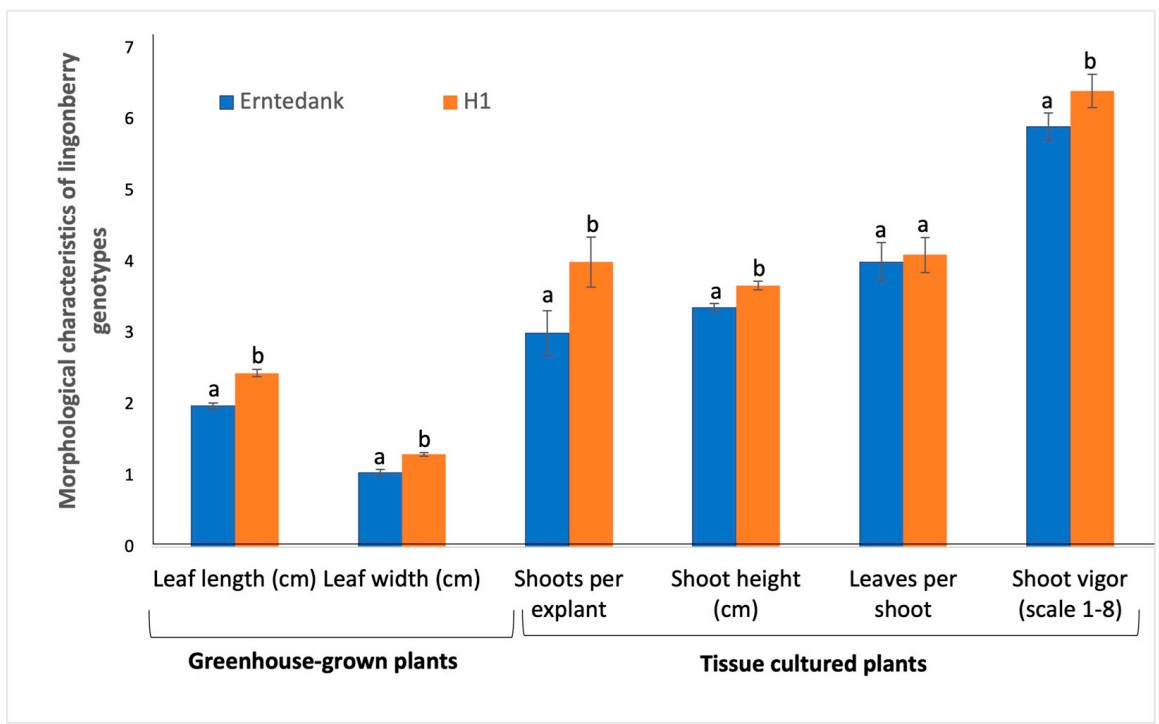

**Figure 2.** Morphological characteristics (means ± standard error) of lingonberry cultivar Erntedank and hybrid H1. Data on leaf length and leaf width were recorded from 2-year-old greenhouse-grown plants. Shoot number per explant, shoot height, leave number per shoot and shoot vigor (scale 1–8; 1 being the poorest, and 8 being the best) were taken from 2-month-old tissue cultures. Bars, within the same group, followed by same letter are not significantly different according to Tukey's range test at *p* = 0.05.

### 3.2. Clonal Fidelity

Clonal fidelity in tissue-cultured lingonberry cultivar Erntedank was assessed using 12 molecular markers including one expressed sequence tag PCR (EST PCR), five expressed sequence tag—simple sequence repeats (EST SSR) and six genomic simple sequence repeats (GSSR). Altogether, 31 monomorphic DNA bands were present from 12 molecular markers, yielding an average of 2.5 bands per primer (Table 2). A representative figure including seven markers is shown in Figure 3.

**Table 2.** List of PCR primers including the primer type, primer name, sequence information, annealing temperature, the number of bands present and the size of amplified alleles in the in vitro propagated shoots and greenhouse-grown plants of lingonberry cultivar Erntedank.

| Primer Type | Primer Name | Primer Sequence | Annealing Temperature | Bands Present (No.) | Size of Amplified Alleles (bp) |
|---|---|---|---|---|---|
| EST PCR | CA21 | F:TCCGATAACCGTTACCAAGC R:TATACAGCGACACGCCAAAA | 54 | 2 | 110, 230 |
| EST SSR | CA23 | F:GAGAGGGTTTCGAGGAGGAG R:GTTTAGAAACGGGACTGTGAGACG | 60 | 2 | 100, 175 |
| EST SSR | CA169 | F:TAGTGGAGGGTTTTGCTTGG R:GTTTATCGAAGCGAAGGTCAAAGA | 54 | 2 | 260, 350 |
| EST SSR | CA421 | F:TCAAATTCAAAGCTCAAAATCAA R:GTTTAAGGATGATCCCGAAGCTCT | 60 | 2 | 175, 250 |
| EST SSR | NA398 | F:TCCTTGCTCCAGTCCTATGC R:GTTTCCTTCCACTCCAAGATGC | 60 | 2 | 145, 200 |
| EST SSR | NA1040 | F:GCAACTCCCAGACTTTCTCC R:GTTTAGTCAGCAGGGTGCACAA | 56 | 3 | 150, 210, 350 |
| GSSR | VCCB3 | F:CCTTCGATCTTGTTCCTTGC R:GTTTGATGCAATTGAGGTGGAGA | 62 | 3 | 125, 270, 300 |
| GSSR | VCCI2 | F:AGGCGTTTTTGAGGCTAACA R:TAAAAGTTCGGCTCGTTTGC | 62 | 3 | 130, 300, 325 |
| GSSR | VCCJ9 | F:GCGAAGAACTTCCGTCAAAA R:GTGAGGGCACAAAGCTCTC | 60 | 3 | 75, 120, 135 |
| GSSR | VCCJ1 | F:CTCATGGGTTCCCATAGACAA R:TGCAGTGAGGCAAAAGATTG | 62 | 3 | 275, 300, 350 |
| GSSR | VCCK4 | F:CCTCCACCCCACTTTCATTA R:GCACACAGGTCCAGTTTTTG | 62 | 3 | 100, 140, 150 |
| GSSR | VCCS10 | F:ATTTGGTGTGAAACCCCTGA R:GTTTGCGGCTATATCCGTGTTTGT | 60 | 3 | 150, 175, 215 |

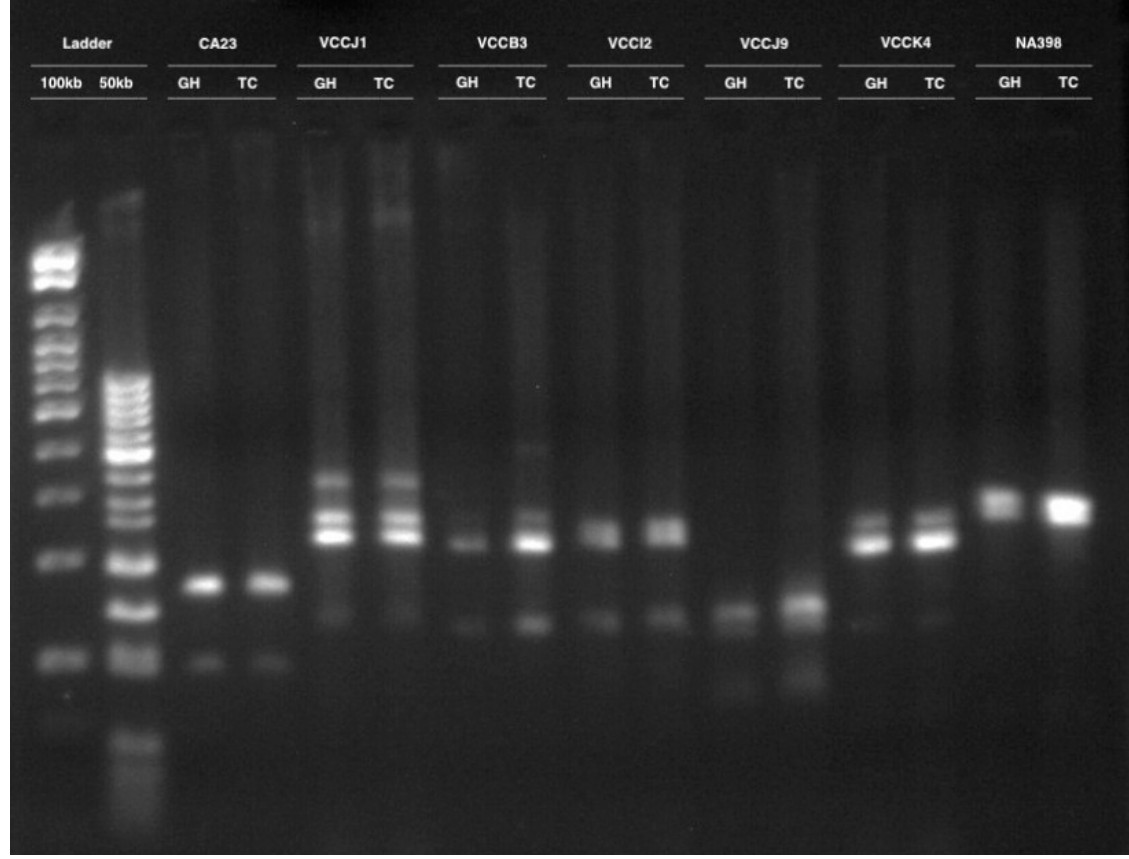

**Figure 3.** Agarose gel image showing DNA banding pattern in greenhouse (GH)- and growth chamber (TC)-grown lingonberry cultivar Erntedank. The first two lanes are the 100 bp and 50 bp DNA ladder. Each of the consecutive two lanes represent the DNA bands with the molecular marker, the first sample being from the greenhouse-grown plants and the second from the tissue-culture shoots in a growth chamber.

### 3.3. DNA Methylation Pattern

DNA methylation profiles were explained based on the polymorphism of the fragments digested with *EcoRI* and one or both of the isoschizomers *MspI/HpaII*, resulting in three lanes. When DNA fragments appeared in all three lanes, they represented non-methylation at the 5′-CCGG-3′ site. *MspI*-specific fragments appeared in the *EcoRI+MspI* and *EcoRI+MspI+HpaII* lanes by the digestion of methylated internal cytosine (5′-CmCGG-3′).

In contrast, *HpaII*-specific fragments appeared in the *EcoRI+HpaII* and *EcoRI+MspI+HpaII* lanes that resulted from cleavage at hemimethylated external cytosine (5′-mCCGG-3′). With *HpaII*-specific fragments, lanes only present on *HpaII* were counted as methylation, as it accounts for internal cytosine methylation [25]. The latter three conditions were considered as methylation at the 5′-CCGG-3′ site. A gel image shows the non-methylated, hemimethylated, and methylated cytosines at the 5′-CCGG-3′ site in Figure 4.

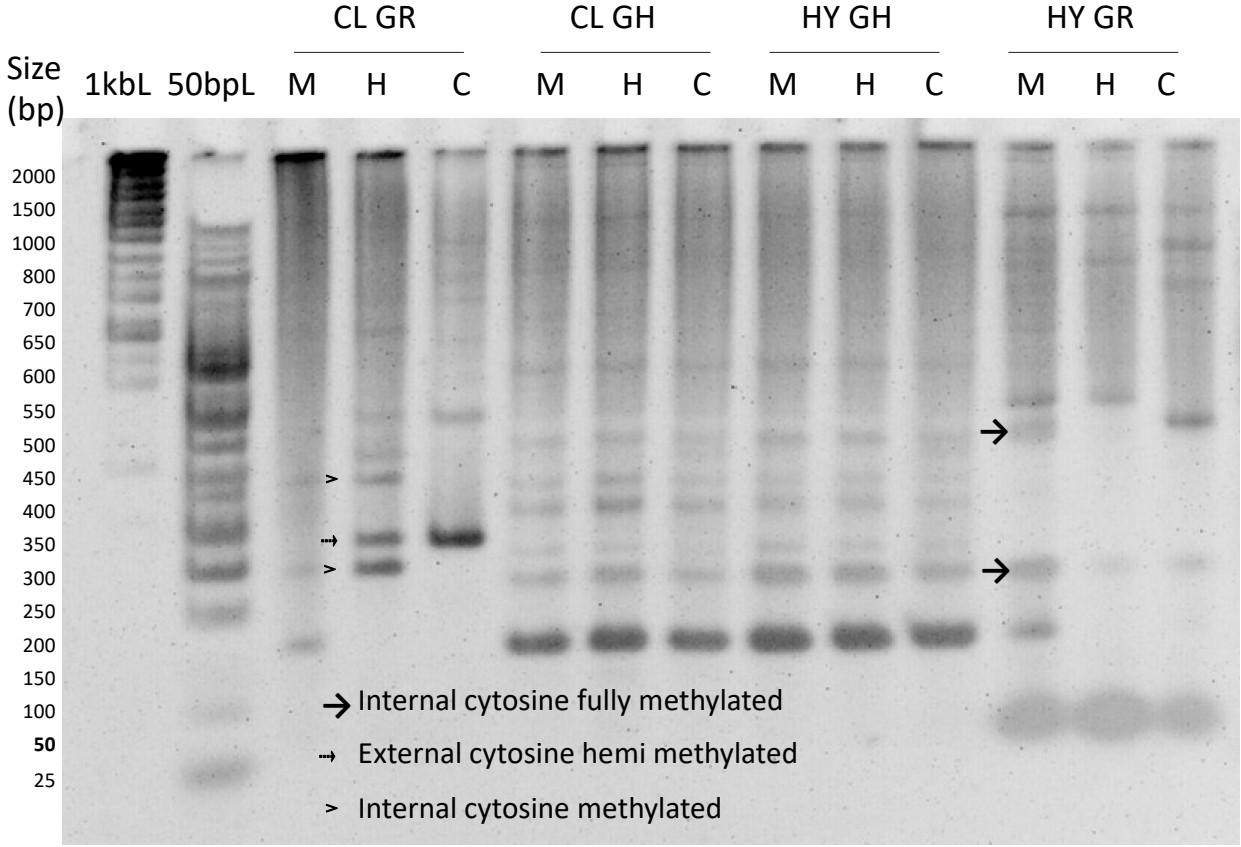

**Figure 4.** Polyacrylamide gel image showing cytosine methylation in the greenhouse (GH) and in vitro grown in semi-solid media in growth chamber (GR) grown lingonberry genotypes, cultivar Erntedank (CL) and a hybrid H1 (HY). CL GR = cultivar growth chamber, CL GH = cultivar greenhouse, HY GR = hybrid growth chamber, and HY GH = hybrid greenhouse. DNA methylation pattern was detected in lingonberry cultivar and hybrid using methylation-sensitive amplification polymorphism (MSAP) assay. Arrows show fully methylated internal cytosine, and arrows with a broken line show hemimethylated external cytosine. The arrowheads represent internal cytosine methylation.

In Erntedank, 344 bands were obtained in the greenhouse-grown plants, and 329 bands were obtained in the tissue-cultured plants (Table 3). In the same genotype, 116 (33.72%) bands were found to be methylated in greenhouse-grown plants, whereas 140 (20%) bands were obtained in tissue-cultured plants. In the hybrid lingonberry, 353 bands were obtained in greenhouse plants, while 364 bands were obtained in tissue-cultured plants. In hybrid lingonberry, 113 (32.01%) of the bands were methylated at the CCGG site, while 160 (43.96%) bands were methylated in tissue cultures. Variation was also observed if the cytosine was fully or hemimethylated. Full methylation ranged between 42.86% to 57.52%, with the highest

observed in greenhouse-grown hybrid lingonberry and the lowest on the tissue-cultured lingonberry cultivar Erntedank (Table 3). Hemimethylation was highest (57.14%) in tissue-cultured lingonberry cultivar and lowest in greenhouse-grown hybrid (42.48%). In general, in vitro grown plants showed higher DNA methylation as compared to the greenhouse-grown plants.

**Table 3.** Cytosine methylation in greenhouse- and growth chamber-grown lingonberries.

| DNA Bands | Erntedank | | Hybrid (H1 = HY GH, HY GR) | |
| --- | --- | --- | --- | --- |
| | Greenhouse | Tissue Culture | Greenhouse | Tissue Culture |
| Type 1 | 228 | 189 | 240 | 204 |
| Type 2 | 54 | 80 | 48 | 86 |
| Type 3 | 46 | 48 | 46 | 50 |
| Type 4 | 16 | 12 | 19 | 24 |
| Total analyzed bands | 344 | 329 | 353 | 364 |
| Total methylated bands | 116 | 140 | 113 | 160 |
| Fully methylated bands | 62 | 60 | 65 | 74 |
| Fully methylated percentage | 53.45% | 42.86% | 57.52% | 46.25% |
| Hemimethylated bands | 54 | 80 | 48 | 86 |
| Hemimethylated percentage | 46.55% | 57.14% | 42.48% | 53.75% |
| MSAP percentage | 33.72% | 42.55% | 32.01% | 43.96% |

## 4. Discussion

Tissue culture is a rapid propagation technique used to propagate transgenic crops and clonally born plants. However, since the tissue culture process bypasses the normal developmental events in the tissue culture microenvironment, it may be stressful for plant tissue, resulting in genetic and epigenetic instabilities. These variations are known as somaclonal variation [22]. Tissue culture-induced variations or their effect in morphological and biochemical response to different stresses have been reported in various plant species [10,27,28]. Morphological characteristics compared between Erntedank and H1 reflected that the selected hybrid H1 had bigger leaves as compared to the cultivar Erntedank. In the tissue cultures also, there were better shoot height, shoot vigor and number of shoots per explant. Phenotypic variations in any organism independent of DNA sequence variation are epigenetic modifications. Although there is no change in the DNA sequence, transcription of the gene is effectively altered by epigenetic factors. Therefore, epigenetic factors are important mediators of gene expression [20]. For the organisms whose genome sequence information is not available, the MSAP technique is widely used to detect epigenetic variations due to DNA methylation in tissue cultures [29,30]. This method has been utilized to determine the epigenetic variation in banana [31], grapevine [32], oil palm [29], blueberry [2] and lingonberry [33].

Variations are a source of novel traits in breeding programs. Phenotypic plasticity is another phenomenon in plant cells that helps the tissues to cope with environmental variation. However, commercial sustainability of the in vitro regeneration systems depends upon the maintenance of genetic integrity. The tissue culture system did not alter the genetic integrity in the lingonberry cultivar in our study. In line with these results, no genetic differences were found in tissue culture-derived plants in blueberry [34], lingonberry [10], or *Jatropha* [35]. However, some genetic variations have been reported in tissue-cultured gerbera [36].

Researchers have reported various levels of cytosine methylation in plants. In our study in lingonberry, cytosine methylation was found to be between 32.01% and 43.96%. However, in tissue-cultured potato, cytosine methylation was found to be very low (0–3.4%) [37]. Methylation in the range of 64.36–67.00% was reported in pepper (*Capsicum* sp.) [38]. Therefore, it can be said that there exists a wide range of variation in the DNA methylation profile in the plant kingdom. Moreover, the effect of micropropagation on cytosine methylation in lingonberries was similar to that in banana [31], orchid [14] and blueberry [2] plants (Table 3), where in vitro conditions resulted in higher cytosine methylation. The organized tissues in the presence of growth regulators dedifferentiate to the undifferentiated mass of totipotent cells called callus and further re-differentiate to

produce plant organs [39]. This alteration in the differentiation status might have resulted in higher cytosine methylation in tissue-cultured lingonberries.

The elusive aspect of epigenetic mechanisms is their variable inheritance. During mitosis cell division, such as in asexual propagation, such variations are frequently heritable, sometimes for multiple generations [39]. However, during sexual reproduction, epigenetic marks are partially reset during meiosis and partially transmitted through meiosis. For example, the epigenetic marks at locus *FLC* affecting the vernalization response are changed during the meiotic process, although transposon methylation is firmly maintained [40]. In another study with a methylation-deficient *Arabidopsis* mutant, it was revealed that methylated CpG are fundamental to epigenetic memory throughout generations [41]. Unlike genetic alleles, epialleles have a tendency to react more frequently to change in the environment, are reversible, and can be retained for a number of generations only [42]

Environmental factors such as exposure to stress change DNA methylation patterns in plants [43]. This adaptive mechanism of plants under altered ecological conditions affects gene expression, including the genes that are involved in the synthesis of biochemicals that have a significant role in abiotic stress tolerance [20]. In an experiment with high-temperature exposure in in vitro conditions, 60% of grapevine somaclones retained the altered DNA methylation pattern even one year after the treatment [32]. In the current study, 16 selective primers revealed lower levels of methylated loci in greenhouse-grown plants than in tissue-cultured plants (Table 3). This trend was similar for both of the genotypes used in the study. In *Oryza sativa*, DNA methylation in the promoter regions of genes has been shown to reduce their expression levels, thus affecting the phenotype [44]. Li et al. [45] also established a relationship between DNA methylation in promoter regions and gene expression, which showed a negative correlation between gene expression and cytosine methylation levels in the 2 kb regions of the promoter [44]. In our experiments, although the location of the DNA methylation in the lingonberry chromosome was not known, if it occurs in the functional DNA region such as a promoter, it could cause significant physiological changes, including those affecting antioxidant properties.

DNA methylation plays a vital role in regulating dedifferentiation and redifferentiation phases in the tissue culture system. In the tissue culture, each plant cell of an explant experiences the medium differentially, and their altered responses result in polymorphism in the methylation pattern [35]. In lingonberry genotypes, higher variation was observed if the cytosine was fully methylated or hemimethylated, which was found irrespective of the percentage of DNA methylation. However, although full methylation was higher in greenhouse-grown plants than in in vitro-grown shoots, the hemimethylated DNA bands were more prominent under in vitro conditions (Table 3). Somaclonal variations (genetic and epigenetic) take place in plants due to environmental stress. MET1, the main cause of methylation, is present in tissue culture plants. The PGRs zeatin and indole-3-butyric acid were used for shoot proliferation in vitro and rooting of microshoots, respectively [10]. The expression of the AUXIN RESPONSE FACTOR3 (ARF3) gene is inhibited by hyper-methylation, giving rise to apical dominance of micropropagated plants, and hypo-methylation enhances expression of the ARF3 gene. In the current investigation, the variable DNA methylation (both hypo- and hyper-methylation) might have been due to a number of factors, including culture conditions, media type, and type and concentration of PGR during lingonberry propagation under ex vitro and in vitro conditions [33]. Under tissue culture conditions, variation in the epigenetic pattern was genotype-specific (Table 3). Similar results have been obtained in blueberry [2], where tissue-cultured blueberry showed higher methylation in a genotype-specific manner. This process may also be linked to alteration in the plant material's hormonal balance and hormone signaling pathway [45,46]. The plant tissue culture process involves the action of PGRs and a complex network of interactions among them. In *Arabidopsis thaliana*, adding indole-3-butyric acid and zeatin in a shoot induction medium resulted in hypo-methylation, which enhanced the expression of the gene encoding ARF3 [47]. Tissue-cultured blueberry in PGR-containing medium exhibited higher cytosine methylation than the cutting or donor counterparts,

respectively [2]. In this line, zeatin (1 mg L$^{-1}$) added in the tissue-cultured media seemed to cause lingonberry's increased DNA methylation.

Lingonberries are characterized as a superfood because of their high antioxidant properties. Several secondary metabolites have antioxidant properties and are regulated by DNA methylation by modulating the expression of key genes involved in this process [48]. Several experiments have shown the involvement of increased DNA methylation to suppress the genes' function, thus resulting in reduced products of those genes. For example, in red sage (*Salvia miltiorrhiza*), *S*-adenosyl methionine (SAM), a donor for DNA methylation, dramatically inhibited accumulation of the phenolic compound [49]. In contrast, the expressions of key genes involved in phenolic acid biosynthesis were downregulated [49], whereas 5-azacytidine, an inhibitor of methylation, significantly enhanced the accumulation of phenolic compounds with a significant upregulation of the key gene expressions involved in phenolic acid biosynthesis [49]. In *Arabidopsis thaliana*, hypomethylation resulted in a substantial increase in the production of a protein involved in growth regulation [50]. CG methylation in *Arabidopsis* is maintained by a conserved METHYLTRANSFERASE1 (*MET1*), a protein homologous to animal *DNMT1* [48,51]. However, methylation in the CHG context is maintained by CHROMOMETHYLASE3 (*CMT3*), and CHH context is maintained by the plant-specific CHROMOMETHYLASE2 (*CMT2*) [51,52]. Analysed expression patterns of *MET1*, *CMT* methyltransferases exhibited higher expression levels in fast-growing calli, and regenerated plants were hypermethylated [52]. In another experiment, genes *MET1* and *CMT3* that code for DNA methylases during somatic embryo formation were found to be upregulated, while genes encoding DNA demethylases were downregulated [51,53]. Therefore, global DNA methylation seemed to affect the transcriptional activity of coding genes, ultimately affecting several physiological processes, including the production of secondary metabolites. In *Arabidopsis thaliana*, a statistical model predicted that 65% of the variance in plant height was the result of DNA methylation [51]. The significance of DNA methylation to the amount of antioxidant production could be particularly important in this medicinally important crop, lingonberry.

## 5. Conclusions

Although no genetic distinction was observed in the lingonberry cultivar Erntedank, variation in DNA methylation patterns in leaf tissues of tissue-cultured and greenhouse-grown lingonberries suggested the effect of in vitro propagation. These variations in turn might affect the expression of the genes involved in vital processes. Given the elusive nature of variation, it would be interesting to evaluate the effects on the vegetative and reproductive stages of mature plants to determine if the tissue culture-induced variation is transient or permanent. This study might be a valuable consideration for the use of commercial micropropagation of lingonberry. Another exciting research direction could be to investigate the effect of epigenetic marks on vital physiological processes, such as the pathways for production of antioxidant compounds.

**Author Contributions:** U.S.: Conducted all of the experiments and wrote the original manuscript draft. A.S.: Helped in conducting experiments. A.U.I.: Co-supervised the research and helped in research planning and execution. S.C.D.: Had roles in conceptualization, supervision, research activity planning and execution, funding acquisition, manuscript writing, and reviewing and editing. All authors have read and agreed to the published version of the manuscript.

**Funding:** This research received no external funding.

**Institutional Review Board Statement:** Not applicable.

**Informed Consent Statement:** Not applicable.

**Data Availability Statement:** Original data presented in the study are included in the main text, and further inquiries can be directed to the corresponding author.

**Acknowledgments:** The authors thank Darryl Martin for his excellent technical help. All of the experiments were performed at St. John's Research and Development Centre (SJRDC), Agriculture and Agri-Food Canada and in the Department of Biology, Memorial University of Newfoundland. This work is an SJRDC contribution.

**Conflicts of Interest:** The authors declare that they have no known competing financial interests or personal relationships that could have appeared to influence the work reported in this paper.

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
