# Peer review of "Exploring Genetic and Epigenetic Changes in Lingonberry Using Molecular Markers: Implications for Clonal Propagation"

_cimb, doi:10.3390/cimb45080397_

Round 1

Reviewer 1 Report

This manuscript describes the assessment of genetic and epigenetic variations in lingonberry under in vitro and greenhouse conditions using molecular markers. My comments are as follows:

Why was the Erntedank cultivar chosen for comparison with the hybrid? If this cultivar is not one of the parents of the hybrid, then what is the idea of comparing morphology? In a pair of randomly selected genotypes, one will outperform the other. In another random pair, the leaves of the cultivar may be bigger than that of the hybrid.

Why was clonal fidelity analysis done only for the cultivar, but not for the hybrid?

Separation of DNA fragments on a Genetic Analyzer would give more accurate results than separation by gel electrophoresis.

L. 131-132. “The first two lanes are the DNA ladder of 100 kb and 50 kb size respectively”. This sentence should be deleted as it is duplicated in the caption to Fig. 2.

L. 188-189. “Morphological characteristics compared between the cultivar and hybrid reflected the hybrid lingonberry have bigger leaves as compared to the cultivar”. This does not mean the advantage of lingonberry hybrids over cultivars in general, since the selected genotypes are not related.

In Materials and Methods, it is necessary to include subsection of statistical analysis.

L. 82, 135, 170, etc. Arabidopsis and in vitro should be italicized.

Fig. 1. What does bar (SD or SE) mean?

Author Response

This manuscript describes the assessment of genetic and epigenetic variations in lingonberry under in vitro and greenhouse conditions using molecular markers. My comments are as follows:

Why was the Erntedank cultivar chosen for comparison with the hybrid? If this cultivar is not one of the parents of the hybrid, then what is the idea of comparing morphology? In a pair of randomly selected genotypes, one will outperform the other. In another random pair, the leaves of the cultivar may be bigger than that of the hybrid.

Response: Thanks for your comments. While Erntedank is a cultivar, the hybrid has been developed at Agriculture and Agri-Food Canada’s St. John’s Research Centre (parental identification is confidential and it is the property of Agriculture and Agri-Food Canada, Government of Canada) The Erntedank and the hybrid were considered as two independent genotypes and thus were compared to see the differences between the two genotypes.

Why was clonal fidelity analysis done only for the cultivar, but not for the hybrid?

Response: Clonal fidelity analysis was done for cultivar Erntedank only and it was assumed that similar results would be for and all H1 tissue culture plants are true-to-type.

Separation of DNA fragments on a Genetic Analyzer would give more accurate results than separation by gel electrophoresis.

Response: It would be, but we used the facility available in our lab.

  1. 131-132. “The first two lanes are the DNA ladder of 100 kb and 50 kb size respectively”. This sentence should be deleted as it is duplicated in the caption to Fig. 2.

Response: According to your suggestions, it has been removed.  

  1. 188-189. “Morphological characteristics compared between the cultivar and hybrid reflected the hybrid lingonberry have bigger leaves as compared to the cultivar.” This does not mean the advantage of lingonberry hybrids over cultivars in general, since the selected genotypes are not related.

Response: The sentence has been revised. It is a comparison between two genotypes.

In Materials and Methods, it is necessary to include subsection of statistical analysis.

Response: Subsection has been added.

  1. 82, 135, 170, etc. Arabidopsis and in vitro should be italicized.

Response: It has been resolved.

Fig. 1. What does bar (SD or SE) mean?

Response: After including the Materials and Method section before the Results section, it is now Fig.2.  The bar represents SE (standard error). Figure 2 legend has been revised.

Reviewer 2 Report

I have carefully read the article entitled "Exploring genetic and epigenetic changes in lingonberry using molecular markers: Implications for clonal propagation". Under in vitro culture, uniformity of planting materials is not achieved due to genetic and epigenetic instabilities and the progeny derived using this technique may not be true-to-type. The present study focuses on the analysing the morphological traits and genetic and epigenetic variations under tissue culture and greenhouse conditions in lingonberry using molecular markers.

The experimental set-up, analysis, and statistical treatment sets have been carefully planned and conducted.

A few clarifications are needed:

  1. The authors had chosen ten plants from each genotype under study to record the morphological data. For the genetic analysis, five plants were randomly selected, for the DNA analysis; I think the selection of more plants would have given better results and clarity in the research.
  2. The authors recorded data on leaf length and leaf width from two-year-old greenhouse-grown plants. Shoot number per explant, shoot height, leave number per shoot and shoot vigor (scale 1-8; 1 being the poorest and 8 being the best) were taken from two-month-old tissue cultures, I think the methodology and the comparison may not be valid.
  3. Broadly speaking, the tissue culture system did not alter the genetic integrity in lingonberry cultivar in the present study. More precisely, no genetic differences were found in tissue culture derived plants in the present study.
  4. No much variation was observed in the fully methylated percentage 53.45% 42.86% and MSAP percentage of 33.72% 42.55% between greenhouse and growth chamber grown lingonberries (tissue culture).
  5. Need clarification on how the total methylation was higher in vitro grown shoots and the fully methylated bands observed were more prominent in the greenhouse grown plants. In contrary, hemi methylated DNA bands were more prominent in tissue culture conditions as compared to greenhouse grown plants.

Hence, my final decision on the article is to accept the manuscript with major revisions.

Author Response

The manuscript describes the evaluation of genetic stability and epigenetic changes during in vitro culture of lingonberry. For this species, in vitro propagation is a good option considering the poor rhizome production in this species and not true-to-type progeny derived from seeds. The authors used three techniques to assess the genetic and epigenetic changes produced by evaluating 12 molecular markers. In general, the manuscript is well presented and written, but some comparisons and results are inconsistent.

Some issues that were found in the manuscript:

Abstract

L16-17 Leaf length, leaf width, shoot number per explant, shoot height, and shoot vigor were significantly higher in hybrid H1 as compared to the cultivar Erntedank.  …Cultivated under in vitro conditions or greenhouse-grown? The idea is not clear.

Response: The sentence has been revised (lines 17-18).

2.1 Morphological characteristics.

-In the section, the age of plants characterized in both cultivation systems is unclear. It is only indicated in the Figure 1 caption.

Response: Now its Figure 2 after including materials and method sections before results sections. This uncleared problem has been resolved in the manuscript (yellow higlighted).

-I understand evaluating the same morphological characteristics

in both systems is impossible. However, it isn’t clear why they only evaluated the leaf characteristics in greenhouse-grown plants. What about height, biomass? In vitro culture, shoot vigor showed significant differences between both materials. What about under greenhouse conditions?

Response: We were unable to take the biomass data as all plants were maintained for other in-depth studies. Under greenhouse conditions, we were able to take leaf length and width and under in vitro condition, we took rest of shoot characteristics and shoot length.

Fig. 1 caption: Values are means? SD or SE? Please, indicate the statistical test used.

Response: Corrections have been done as per suggestions (lines 272-274; 289; 293-294).

2.2. Clonal fidelity

-Why only the Erntedank cv plants were evaluated for clonal fidelity? What happened with H1 plants?. In contrast, H1 plants were considered for methylation analyses.

Response: Clonal fidelity analysis was done for cultivar Erntedank only and it was assumed that similar results would be for and all H1 tissue culture plants are true-to-type as for other previous results with different genotypes from the same lab. However, DNA methylation was assumed to be genotype dependent (also reported by the same group members from same lab).

-I'm sorry, but I don't find any sense in saying molecular markers were used to evaluate the clonal fidelity and then show results in gel for only seven. Why only seven? There is no limitation of space in electronic journals.

Response: Some representative results from some of the shoots of in vitro- and ex vitro-grown plants are presented in the figure 3. All results are available in Table 2.

Table 1. Size of amplified alleles (bp) column: There needs to be clarified if the numbers indicated in this column are for both types of plants. If so, in some cases, the numbers of fragments indicated are not in line with that observed in gel (Fig. 2), v.g.: VCCB3.

Response: After including the Materials and Methods section before the Results section, Fig. 2 has been replaced by Fig. 3. In the picture, the bands are little bit fainted in one lane that might be due to less concentrated DNA (greenhouse sample). However, the presence bands was confirmed by repeated visual observation.

Reviewer 3 Report

The manuscript describes the evaluation of genetic stability and epigenetic changes during in vitro culture of lingonberry. For this species, in vitro propagation is a good option considering the poor rhizome production in this species and not true-to-type progeny derived from seeds. The authors used three techniques to assess the genetic and epigenetic changes produced by evaluating 12 molecular markers. In general, the manuscript is well presented and written, but some comparisons and results are inconsistent.

Some issues that were found in the manuscript:

Abstract

L16-17 Leaf length, leaf 16 width, shoot number per explant, shoot height, and shoot vigor were significantly higher in hybrid 17 H1 as compared to the cultivar Erntedank.  …Cultivated under in vitro conditions or greenhouse-grown? The idea is not clear.

2.1 Morphological characteristics.

-In the section, the age of plants characterized in both cultivation systems is unclear. It is only indicated in the Figure 1 caption.

-I understand evaluating the same morphological characteristics

in both systems is impossible. However, it isn’t clear why they only evaluated the leaf characteristics in greenhouse-grown plants. What about height, biomass? In vitro culture, shoot vigor showed significant differences between both materials. What about under greenhouse conditions?

Fig. 1 caption: Values are means? SD or SE? Please, indicate the statistical test used.

2.2. Clonal fidelity

-Why only the Erntedank cv plants were evaluated for clonal fidelity? What happened with H1 plants?. In contrast, H1 plants were considered for methylation analyses.

-I'm sorry, but I don't find any sense in saying 12 molecular markers were used to evaluate the clonal fidelity and then show results in gel for only seven. Why only seven? There is no limitation of space in electronic journals.

Table 1. Size of amplified alleles (bp) column: There needs to be clarified if the numbers indicated in this column are for both types of plants. If so, in some cases, the numbers of fragments indicated are not in line with that observed in gel (Fig. 2), v.g.: VCCB3.

Author Response

I have carefully read the article entitled "Exploring genetic and epigenetic changes in lingonberry using molecular markers: Implications for clonal propagation". Under in vitro culture, uniformity of planting materials is not achieved due to genetic and epigenetic instabilities and the progeny derived using this technique may not be true-to-type. The present study focuses on the analysing the morphological traits and genetic and epigenetic variations under tissue culture and greenhouse conditions in lingonberry using molecular markers.

The experimental set-up, analysis, and statistical treatment sets have been carefully planned and conducted.

Response: Thanks for your comments.

A few clarifications are needed:

  1. The authors had chosen ten plants from each genotype under study to record the morphological data. For the genetic analysis, five plants were randomly selected, for the DNA analysis; I think the selection of more plants would have given better results and clarity in the research.

Response: Generally 5-10 random plants can be taken for data collection. We used 10 plants for morphological study and 5 plants for DNA analysis to make the experiment cost effective. Although DNA analysis if relatively expensive.

  1. The authors recorded data on leaf length and leaf width from two-year-old greenhouse-grown plants. Shoot number per explant, shoot height, leave number per shoot and shoot vigor (scale 1-8; 1 being the poorest and 8 being the best) were taken from two-month-old tissue cultures, I think the methodology and the comparison may not be valid.

Response: Data on leaf length and leaf width were from greenhouse-grown plants for both genotypes. Similarly, shoot number per explant, shoot height, leave number per shoot and shoot vigor for both genotypes were done under in vitro condition, and hence, the methodology is correct and the statistical comparison between two genotypes is valid.

  1. Broadly speaking, the tissue culture system did not alter the genetic integrity in lingonberry cultivar in the present study. More precisely, no genetic differences were found in tissue culture derived plants in the present study.

Response: Yes, genetically they are same although epigenetic variation is there.

  1. No much variation was observed in the fully methylated percentage 53.45% 42.86% and MSAP percentage of 33.72% 42.55% between greenhouse and growth chamber grown lingonberries (tissue culture).

Response: Thanks for your comments. Our results are explained in Results and Discussion sections in detail.

  1. Need clarification on how the total methylation was higher in vitro grown shoots and the fully methylated bands observed were more prominent in the greenhouse grown plants. In contrary, hemi methylated DNA bands were more prominent in tissue culture conditions as compared to greenhouse grown plants.

Response: The manuscript has been revised as was suggested (lines 418-429.

Hence, my final decision on the article is to accept the manuscript with major revisions.

Response: Thank you for your thoughtful and valuable suggestion.

Round 2

Reviewer 1 Report

The manuscript has been improved and may be published.

Reviewer 2 Report

Authors have made significant improvement in the manuscript as per the suggestions of  the reviewers.  The manuscript therefore can be accepted for publication.

Reviewer 3 Report

In this second version of the manuscript, the authors have addressed all the issues on the manuscript done by the reviewers. At this point, I have no more comments. The manuscript is ready for publication.